# Customized Subgraph Selection and Encoding for Drug-drug Interaction Prediction

**Haotong Du**[1]  **Quanming Yao**[2]  **Juzheng Zhang**[2]  **Yang Liu**[1]  **Zhen Wang**[1][†]
[1]Northwestern Polytechnical University  [2]Tsinghua University
duhaotong@mail.nwpu.edu.cn  w-zhen@nwpu.edu.cn
qyaoaa@tsinghua.edu.cn  {juzhengzh00,yangliuyh}@gmail.com

## Abstract

Subgraph-based methods have proven to be effective and interpretable in predicting drug-drug interactions (DDIs), which are essential for medical practice and drug development. Subgraph selection and encoding are critical stages in these methods, yet customizing these components remains underexplored due to the high cost of manual adjustments. In this study, inspired by the success of neural architecture search (NAS), we propose a method to search for data-specific components within subgraph-based frameworks. Specifically, we introduce extensive subgraph selection and encoding spaces that account for the diverse contexts of drug interactions in DDI prediction. To address the challenge of large search spaces and high sampling costs, we design a relaxation mechanism that uses an approximation strategy to efficiently explore optimal subgraph configurations. This approach allows for robust exploration of the search space. Extensive experiments demonstrate the effectiveness and superiority of the proposed method, with the discovered subgraphs and encoding functions highlighting the model's adaptability.

## 1 Introduction

Precise prediction of drug-drug interactions (DDIs) is essential in biomedicine and healthcare research [1]. Drug combination therapy [2] can enhance treatment effectiveness for certain diseases; however, it also increases the risk of adverse drug reactions, potentially threatening patient safety [3]. Identifying DDIs through laboratory experiments is both costly and time-consuming [4, 5]. With the success of deep learning, researchers have increasingly explored computational methods for DDI prediction. Early approaches primarily relied on molecular fingerprint information [6] or hand-engineered features [7], often neglecting the pre-existing interaction properties between drugs.

Considering drugs as nodes and their interactions as edges, DDI prediction can be framed as a multi-relational link prediction problem within the constructed drug interaction network. Recent advancements in graph neural networks (GNNs) [8, 9, 10, 11] have consistently achieved superior performance in this task. Specifically, subgraph-based methods, such as SumGNN [12], EmerGNN [13], and KnowDDI [14], have shown promising results by selecting subgraphs around query edges and applying sophisticated encoding functions (message passing functions) to represent these subgraphs, Such methods transform the multi-relational link prediction task into a multi-type subgraph classification problem. Figure 1 illustrates the pipeline of subgraph-based methods.

However, due to the dense nature [15, 16] of drug interaction networks and their complex interaction semantics [17], existing hand-designed subgraph methods often fail to capture the nuanced but crucial information across different data inputs. In the initial phase of the reasoning pipeline, the subgraph sampler must have the capability to customize the selection of drug subgraphs for different queries, thereby ensuring precise contextualization of the reasoning evidence.

---

[†]Corresponding author.

38th Conference on Neural Information Processing Systems (NeurIPS 2024).

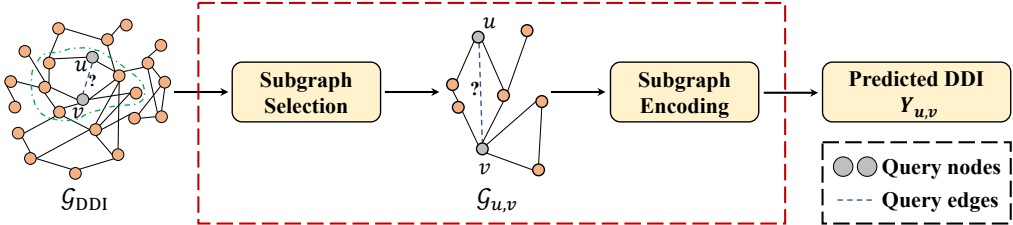

Figure 1: The pipeline of subgraph-based methods includes subgraph selection and subgraph encoding. In this work, we focus specifically on searching for components within the red-dotted lines.

Without customized subgraph selection, SumGNN samples subgraphs using a fixed subgraph range $k$, selecting the $k$-hop neighbors of each drug as associated subgraphs for predicting drug-drug interactions (DDIs). This coarse-grained approach is straightforward and easy to implement, but it may introduce noise or omit valuable information needed to reason about diverse drug pair interactions.

In terms of encoding process, the encoding function must be capable of modeling a wide variety of drug interactions within the drug interaction network. Real-world drug interactions exhibit complex mechanisms, for instance, metabolism-based interactions display asymmetric semantic patterns, whereas phenotype-based interactions are symmetric. Manually designed encoding functions are limited in their ability to accommodate both types of distinct semantic patterns simultaneously [23]. Therefore, designing a customized and data-adaptive subgraph-based pipeline is essential for effective DDI prediction.

Table 1: Comparing with existing methods."-" represents not applicable.

| Method | Fine-grained Subgraph Selection | Data-specific Encoding Function |
|---|---|---|
| SEAL [18] | ✗ | ✗ |
| GraIL [19] | ✗ | ✗ |
| SumGNN [12] | ✗ | ✗ |
| SNRI [20] | ✗ | ✗ |
| KnowDDI [14] | ✓ | - |
| MR-GNAS [21] | - | ✓ |
| AutoGEL [22] | - | ✓ |
| **CSSE-DDI** | ✓ | ✓ |

Neural architecture search (NAS) [24, 25] has achieved remarkable success in designing data-specific models, often surpassing architectures crafted by human experts in various fields, such as computer vision [26], graph neural network [27], and knowledge graph learning [23]. However, effectively selecting suitable subgraphs from the vast space of candidates and efficiently optimizing the joint search process of subgraph selection and encoding remain open challenges.

In this paper, we leverage NAS to search for data-specific components in the subgraph-based pipeline. Specifically, we design search spaces for pipeline components, including subgraph selection and encoding spaces, to capture various drug interaction patterns. To enable efficient exploration of the extensive subgraph selection space, we introduce a relaxation mechanism that continuously selects subgraphs in a structured manner. Additionally, we propose a subgraph representation approximation strategy to reduce the high cost of explicit subgraph sampling, enabling efficient and robust search. Compared with existing methods in Table 1, our proposed **C**ustomized **S**ubgraph **S**election and **E**ncoding for **D**rug-**D**rug **I**nteraction prediction (**CSSE-DDI**) achieves fine-grained subgrpah selection and data-specific encoding functions, providing an efficient and precise method for drug interaction prediction. Our main contributions are summarized as follows:

- We present CSSE-DDI, a searchable framework for DDI prediction that adaptively customizes the subgraph selection and encoding processes. To the best of our knowledge, this is the first application of NAS techniques to tailor an adaptive subgraph-based pipeline for the DDI prediction task.

- We construct expressive search spaces to ensure precise capture of evidence for drug interaction prediction. Additionally, we devise a relaxation mechanism to transform the discrete subgraph selection space into a continuous form, enabling differentiable search. Simultaneously, we apply a subgraph representation approximation strategy to mitigate the inefficiencies of explicit subgraph sampling, thereby accelerating the search process.

- Extensive experiments on benchmark datasets demonstrate that our method, which searches for customized pipelines, achieves superior performance compared to hand-designed methods. Additionally, our approach effectively captures the underlying biological mechanisms of drug-drug interactions.

## 2 Related Works

**Subgraph-based Link Prediction**    Recently, subgraph-based methods [18, 19, 28] have emerged as a promising approach, showing superior performance in link prediction tasks. Different from canonical GNNs, subgraph-based method extracts a subgraph patch for each training and test query, learning a representation of the extracted patch for final prediction, as illustrated in Figure 1.

Existing works has primarily focused on designing more informative subgraphs and more expressive encoding functions. However, they do not take into account customizing these components to deal with various data. Specifically, in terms of subgraph sampler, current approaches lack fine-grained and adaptive extraction for different query subgraphs. While PS2 [29] demonstrates the effectiveness of identifying optimal subgraphs for each edge in homogeneous graph link prediction, there is no comparable work in multi-relational graph link prediction. In dense DDI networks, fine-grained identification of subgraph for different queries is even more crucial.

As for the encoding function, existing works overlook the importance of data-specific encoding, which has been emphasized in recent literature [30, 27]. Customized encoding functions are especially advantageous for drug interaction networks with complex and diverse interactions.

**GNN-based DDI Prediction**    Recently, there has been growing interest in applying GNNs for DDI prediction [8, 9]. However, these works execute message-passing functions over the entire graph, which limits their ability to capture explicit local evidence for specific query drug pairs and lack interpretability. In contrast, subgraph-based DDI prediction methods [12, 63, 13, 14] transform the multi-relational link prediction problem into a subgraph classification problem by extracting subgraphs around query nodes, achieving strong performance. Nevertheless, these works use the same subgraph extraction strategy for all queries and rely on a fixed message-passing function to handle complex DDIs, which limits their flexibility and adaptivity in dense DDI networks.

**Graph Neural Architecture Search**    Graph neural architecture search (GNAS) [31] aims to find high-performing GNN architectures using NAS techniques. Recent studies [30, 27] have explored GNAS to create more expressive GNN models across various tasks. AutoDDI [32], for instance, automatically designs GNN architectures to learn molecular graph representations of drugs for DDI prediction. However, research on optimizing graph sampling for GNAS remains limited due to the diversity of graph-structured data.

Regarding search strategy, early approaches explores the search space using reinforcement learning [33] or evolutionary algorithms [34], which is highly inefficient. One-shot approaches [35] instead construct an over-parameterized network (supernet) and optimize it using gradient descent, leveraging continuous relaxation of the search space to improve search efficiency. The recently proposed few-shot NAS paradigm [36] further enhances supernet evaluation consistency by generating multiple sub-supernets.

## 3 Proposed Method

### 3.1 Problem Formulation

Given a set of drugs $\mathcal{V}$ and interaction relations $\mathcal{R}$ among them, the drug interaction network is denoted as $\mathcal{G}_{\text{DDI}} = \{(u, r, v) \mid u, v \in \mathcal{V}, r \in \mathcal{R}\}$, with each tuple $(u, r, v)$ describes an interaction between drug $u$ and drug $v$. Consequently, drug-drug interaction (DDI) prediction can be framed a multi-relational link prediction task within the drug interaction network $\mathcal{G}_{\text{DDI}}$. The objective is to predict the types of interactions between two given drug nodes, which can be denoted as a query $(u, ?, v)$, i.e., given the query drug-pair entities $u$ and $v$, to determine the interaction $r$ that makes $(u, r, v)$ valid.

Moreover, instead of directly predicting on the entire graph $\mathcal{G}_{\text{DDI}}$, subgraph-based methods decouple the prediction process into two stages: (1) selecting a query-specific subgraph and (2) encoding the subgraph to predict interactions, as shown in Figure 1. The prediction pipeline then becomes

$$\mathcal{G}_{\text{DDI}} \xmapsto{\text{Selection},(u,v)} \mathcal{G}_{u,v} \xmapsto{\text{Encoding}} \boldsymbol{Y}_{u,v}, \tag{1}$$

where the sampler selects a subgraph $G_{u,v}$ conditioned on the given query $(u, ?, v)$. Using this subgraph $G_{u,v}$, the encoding function produces the final predictions $\boldsymbol{Y}_{u,v}$.

Building on previous analysis and existing research, and inspired by NAS, we propose to search for data-adaptive subgraph selection and encoding components to obtain a customized subgraph

pipeline. In Section 3.2, we first introduce the well-designed subgraph selection and encoding spaces to ensure comprehensive coverage of cricual information in various drug interaction networks. Further, in Section 3.3 we present a subgraph relaxation strategy and approximation mechanisms for subgraph representations to facilitate efficient differentiable search. Finally, we develop a robust search algorithm to address the customized search problem with stability and precision.

## 3.2 Search Space
### 3.2.1 Subgraph Selection Space

In practice, subgraph-based methods define the drug-pair subgraph between drug pairs as the union or interaction of $k$-hop ego-network [2] of query drugs. Here, $k$ is a key hyperparameter that determines the range of message propagation aggregated by the central node. Selecting $k$ is crucial to model performance, as it dictates whether the model has access to high-quality evidence context for accurate prediction.

Prior works [12, 37] typically utilize a fixed hyperparameter for all drug pairs, i.e., selecting the union of a fixed $k$-hop ego-network for arbitrary queries. Nevertheless, this approach can lead to an imprecise collection of evidence for interaction reasoning, potentially undermining the reasoning process due to missing critical information or the inclusion of excessive irrelevant information.

Based on the above analysis, we define a drug-pair subgraph selction space containing a range of subgraphs of different sizes for a given query $(u, v)$:

$$\mathcal{S}_{u,v} = \{\mathcal{G}_{u,v}^{i,j} \mid 1 \leq i, j \leq \eta\}, \tag{2}$$

where $\mathcal{G}_{u,v}^{i,j}$ is generated by taking the union of the $i$-hop ego-network of node $u$ and the $j$-hop ego-network of node $v$, i.e., $\mathcal{G}_{u,v}^{i,j} = \{z \in \mathcal{V} \mid z \in (u \cup \mathcal{N}_i(u) \cup v \cup \mathcal{N}_j(v))\}$, where $\mathcal{N}_i(u)$ and $\mathcal{N}_j(v)$ are the $i$-hop and the $j$-hop neighbors of $u$ and $v$, respectively. The threshold $\eta$ constrains the maximum subgraph range.

Since each drug-pair has a specific subgraph selection space, the overall size of space in the entire graph is $\eta^{2|\mathcal{E}|}$, where $|\mathcal{E}|$ represents the number of edges in the drug interaction network. A larger $|\mathcal{E}|$ result in a subgraph selection space that grows exponentially with the number of edges. Therefore, efficiently searching for the optimal subgraph configurations for different queries is challenging.

### 3.2.2 Subgraph Encoding Space

For the automated design of the subgraph encoding function, we first adopt a unified message passing framework [21, 38] comprising several key modules: the message-computing function MES, the aggregation function AGG, the combination function COM, and the activation function ACT, as follows:

$$
\begin{aligned}
&\texttt{step 1: } \mathbf{m}_u \leftarrow \texttt{AGG}(\texttt{MES}(\mathbf{h}_v, \mathbf{h}_{r(u,v)})_{v \in \mathcal{N}_1(u)}), \\
&\texttt{step 2: } \mathbf{h}_u \leftarrow \texttt{ACT}(\texttt{COM}(\mathbf{h}_u, \mathbf{m}_u)),
\end{aligned}
\tag{3}
$$

where $\mathbf{h}_u \in \mathbb{R}^d$ and $\mathbf{h}_r \in \mathbb{R}^d$ represent the embeddings of node $u$ and interaction $r$, respectively, and $\mathbf{m}_u$ is the intermediate message representation of $u$ aggregated from its neighborhood $\mathcal{N}_1(u)$.

A substantial amount of literature [39, 40, 41] has focused on manually designing these modules to improve performance. However, such encoding functions are inflexible for handling diverse interaction patterns across different drug interaction network. For example, interactions in DrugBank [42] describe how one drug affects the metabolism of another one. The excretion of Acamprosate, for instance, may be decreased when combined with Acetylsalicylic acid (Aspirin). Such interaction pattern is asymmetric, meaning $r(x, y) \nRightarrow r(y, x)$. Conversely, interactions in the TWOSIDES dataset [43] are primarily at the phenotypic level, such as headache or pain in throat, representing symmetric patterns where $r(x, y) \Rightarrow r(y, x)$. These two relational semantics are distinctly different, and existing hand-designed encoding functions struggle to capture such diverse semantics effectively [44, 23].

Here, we aim to perform an adaptive searching for the encoding function in the context of drug interaction prediction. Based on the framework presented in Eq. (3), we design an expressive subgraph encoding space with a set of candidate operations. Detailed explanations of these modules can be found in the Appendix A.1.

---

[2] A $k$-hop ego-network of a node consists of the node and its $k$-hop neighbors.

After encoding the subgraph $\mathcal{G}_{u,v}$, we obtain the representation $\mathbf{z}_{u,v}$ of the input subgraph $\mathcal{G}_{u,v}$. The predictor then maps the representation $\mathbf{z}_{u,v}$ to the probability logits for different interactions between drug pairs, i.e., $y_{u,v} = \mathbf{W}_{\text{pred}}\mathbf{z}_{u,v}$, where $\mathbf{W}_{\text{pred}} \in \mathbb{R}^{2d \times |\mathcal{R}|}$ is the parameter of the predictor.

### 3.3 Search Strategy

#### 3.3.1 Search Problem

Based on the well-designed search space described above, we formulate a bi-level optimization problem to adaptively search for the optimal configuration of subgraph-based pipelines.

**Definition 1** (Customized Subgraph-based Pipeline Search Problem). *Let $\mathcal{A}$ denote the subgraph encoding space, $\mathcal{S}_{u,v}$ represent the subgraph selection space for the query $(u, v)$, $\boldsymbol{\alpha}$ be a candidate encoding function in $\mathcal{A}$, $\mathbf{W}$ represent the parameters of a model from the search space, and $\mathbf{W}^*(\mathcal{G}_{u,v}; \boldsymbol{\alpha})$ denote the trained operation parameters. Let $\mathcal{D}_{\text{tra}}$ and $\mathcal{D}_{\text{val}}$ denote the training and validation sets, respectively. The search problem is formulated as follows:*

$$\arg\max_{\boldsymbol{\alpha} \in \mathcal{A}, \mathcal{G}_{u,v} \in \mathcal{S}_{u,v}} \sum_{(u,r,v) \in \mathcal{D}_{\text{val}}} \mathcal{M}(\mathbf{W}^*(\mathcal{G}_{u,v}; \boldsymbol{\alpha}); \mathcal{G}_{u,v}; \boldsymbol{\alpha}), \tag{4}$$

$$s.t.\ \mathbf{W}^*(\mathcal{G}_{u,v}; \boldsymbol{\alpha}) = \arg\min_{\mathbf{W}} \sum_{(u,r,v) \in \mathcal{D}_{\text{tra}}} \mathcal{L}(\mathbf{W}; \mathcal{G}_{u,v}; \boldsymbol{\alpha}), \tag{5}$$

*where the classification loss $\mathcal{L}$ is minimized for all interactions, while the performance measurement $\mathcal{M}$ is expected to be maximized.*

In this work, we adopt the differentiable search paradigm [45] to solve the bi-level optimization problem, which is widely used in recent NAS literature [46] and enables efficient exploration of the search space. Nevertheless, our proposed subgraph selection space poses two technical challenges: **First**, we cannot directly apply relaxation strategies, which is a prerequisite for differentiable NAS methods, to make the discrete selection space continuous. This limitation arises because different subgraphs in the selection space contain diverse nodes and edges, making it challenging to design a relaxation function that unifies subgraphs of varying sizes. **Second**, to enable searching within the subgraph selection space, we would need to first generate all subgraphs in the space. However, sampling such a large number of subgraphs is computationally intractable.

To address these challenges, we design a subgraph selection space relaxation mechanism in Section 3.3.2 . Additionally, we introduce an intuitive subgraph representation approximation strategy in Section 3.3.3 to reduce the high costs associated with explicit sampling.

#### 3.3.2 Relaxation of Subgraph Selection Space

Technically, as in existing NAS works [45, 47], one typically needs to relax the search space into continuous form to enable effective backpropagation training. However, for the subgraph selection space, the traditional continuous relaxation strategy is not directly applicable due to the structural mismatch between graphs and vectors.

To address this, we first utilize encoding function $f(\cdot)$ to encode subgraphs with different scopes. This approach provides all subgraphs with representations of the same dimension, making it feasible to implement a relaxation strategy. Additionally, inspired by the reparameterization trick [48], we adopt the Gumbel-Softmax function to facilitate differentiable learning over a discrete space:

$$\hat{\mathbf{z}}_{u,v}^{i,j} = \sum_{1 \leq i,j \leq \eta} \frac{\exp(\log(g(f(\mathcal{G}_{u,v}^{i,j})) + \mathbf{G}_{i,j})/\tau)}{\sum_{i',j'=1}^{\eta} \exp(\log(g(f(\mathcal{G}_{u,v}^{i',j'})) + \mathbf{G}_{i',j'})/\tau)} f(\mathcal{G}_{u,v}^{i,j}), \tag{6}$$

where $g(\cdot)$ scores the subgraph representations using multiple linear layers, $\mathbf{G}_{i,j} = -\log(-\log(\mathbf{U}_{i,j}))$ is the Gumbel random variable, $\mathbf{U}_{i,j}$ is a uniform random variable, and $\tau$ is the temperature parameter controlling sharpness. $\hat{\mathbf{z}}_{u,v}^{i,j}$ is the mixed selection operation of subgraph $\mathcal{G}_{u,v}^{i,j}$ used to optimize searching process.

#### 3.3.3 Subgraph Representation Approximation Strategy

To solving the optimization problem as Eq. (4) and (5), we need to explicitly sample all the candidate subgraphs within the subgraph selection space $\mathcal{S}_{u,v}$ for each query. However, one of the most challenging aspects of subgraph-based approaches is their inefficient subgraph sampling process [49, 50, 51].

Upon examining our subgraph selection space, we observe that all subgraphs are generated by combining multi-hop ego-networks of the target nodes, encompassing multiple neighborhood hops. Inspired by the $k$-subtree extractor [52], we apply an encoding function to the entire graph and use the resulting node representations of $u$ and $v$ as the ego-network representations for these nodes. The representation of the drug pair can then be obtained by concatenating the ego-network representations of $u$ and $v$. Formally, if we denote by $f(\mathcal{G}_{\text{DDI}}, u, i)$ the $i$-layer hidden representation of node $u$ produced by encoding function applied to $\mathcal{G}_{\text{DDI}}$, then

$$f(\mathcal{G}_{u,v}^{i,j}) \approx \texttt{CONCAT}(f(\mathcal{G}_{\text{DDI}}, u, i), f(\mathcal{G}_{\text{DDI}}, v, j)), \tag{7}$$

The $k$-subtree extractor represents the $k$-subtree structure rooted at a given node, which mirrors the structure as the $k$-hop ego-network. This approximation strategy only requires executing the encoding function on the entire drug interaction network, thereby efficiently yielding subgraph representations of varying scopes, which significantly improves the efficiency in solving the bi-level optimization problem.

### 3.3.4 Robust Search Algorithm

Using the proposed subgraph selection relaxation mechanism, we can transform the overall discrete search space in Definition. 1 into a continuous form, allowing the search problem to be solved by the one-shot NAS paradigm. Additionally, our subgraph representation approximation strategy efficiently obtains subgraph representations and reduces search costs

Following [53], we adopt the single path one-shot training strategy (SPOS) [54] to reduce the computational cost of supernet training. However, the one-shot approach [55, 56, 45], i.e., using the same supernet parameters $\mathbf{W}$ for all architectures, can decrease the consistency between the supernet's performance estimation and the ground-truth performance [57]. Inspired by few-shot NAS [36], we propose a message-aware partitioned supernet training strategy to mitigate the coupling effect of different message-computing operators [58]. By partitioning the superent to form sub-supernets based on the type of message-computing function, this strategy improves the consistency and accuracy of supernet, enabling the search algorithm more stable and robust. Algorithm 1 delineates the full procedure, with further details provided in Appendix A.2.

---

**Algorithm 1:** The search algorithm of CSSE-DDI.

---

**Input:** Supernet $\mathcal{S}$, number of partitions based on message computing function categories $M$ ($M = 4$), subsupernet $\mathcal{S}_i, (i = 1, \cdots, M)$.

    `// supernet training phase`

1 Train $\mathcal{S}$ by continuously sampling a single path until convergence;

    `// supernet partition phase`

2 Partition $\mathcal{S}$ into $M$ sub-supernets $\mathcal{S}_1, \cdots, \mathcal{S}_M$;

    `// sub-supernet training phase`

3 **forall** $i = 1, \cdots, M$ **do**

4     Initialize $\mathcal{S}_i$ with weights transferred from $\mathcal{S}$;

5     Train $\mathcal{S}_i$ by continuously sampling a single path until convergence;

6 **end**

    `// searching phase`

7 Search the optimal encoding function from sub-supernets $\mathcal{S}_1, \cdots, \mathcal{S}_M$ on validation data by natural gradient descent;

8 Select the optimal subgraphs from sub-supernets $\mathcal{S}_1, \cdots, \mathcal{S}_M$ on validation data by preserving the subgraphs with the largest probabilities;

---

### 3.4 Comparison with Existing Works

While many works [12, 13, 14] have explored DDI prediction using subgraph-based methods, our approach introduces two significant advancements. First, to the best of our knowledge, our method (CSSE-DDI) is the first to customize the subgraph selection and encoding processes specifically for subgraph-based DDI prediction. In contrast, previous methods rely on fixed subgraph selection strategy to sample subgraphs and employ hand-designed functions for encoding, as summarized in Table 1. Consequently, our method can adapt data-specific components within subgraph-based pipelines, outperforming existing methods in both performance and efficiency (Section 4.2). Moreover, our approach not only selects fine-grained drug-pair subgraphs that enhance interpretability through potential pharmacokinetic and metabolic concepts (Section 4.6.1), but also searches for data-specific encoding functions that accurately capture the semantic features of drug interactions (Section 4.6.2).

# 4 Experiments

## 4.1 Experimental Setup

**Datasets**   Experiments are conducted on two public benchmark DDI datasets: DrugBank [42] and TWOSIDES [43]. Detailed descriptions of these datasets are presented in Appendix B.1.

**Experimental Settings**   Following [13], we examine two DDI prediction task settings: S0 and S1. Let the drug pairs for DDI prediction be denoted as $(u, v)$. In the S0 setting, both drug nodes $u$ and $v$ are present in the known DDI graph. Existing DDI prediction methods are typically evaluated in this setting. In contrast, the S1 setting involves a pair (u, v) where one drug is known and the other is a novel drug not represented in the known DDI graph. This scenario highlights the critical need for DDI predictions involving new drugs in real-world applications.

**Evaluation Metric**   We follow [12] to evaluate our method. For the DrugBank dataset, where each drug pair contains only one interaction, we use the following metrics: F1 Score, Accuracy and Cohen's $\kappa$. For the TWOSIDES dataset, where multiple interactions may exist between a pair of drugs, we consider the following metrics: ROC-AUC, PR-AUC and AP@50. Additional details are provided in Appendix B.2.

**Baselines**   We compare CSSE-DDI with the following representative DDI prediction method: (i) GNN-based methods include Decagon [8], GAT [59], SkipGNN [9], CompGCN [60], ACDGNN [61], and TransFOL [62]. (ii) Subgraph-based methods include SEAL [18], GraIL [19], SumGNN [12], SNRI [20], KnowDDI [37] and LaGAT [63]. (iii) NAS-based method include MR-GNAS [21], and AutoGEL [22].

We also compare our method with two variants, including CSSE-DDI-FS and CSSE-DDI-FF. The configurations of these variants are as follows: (i) **CSSE-DDI-FS**: This variant omits fine-grained subgraph selection for each query, using fixed k-layer drug node representations to generate the subgraph representation. (ii) **CSSE-DDI-FF**: This variant does not search for the encoding function, instead using a fixed encoding function backbone to capture semantic and topological features in the drug interaction network. In this case, we employ a 3-layer CompGCN model as the backbone. For all baselines, we obtain the results by rerunning the released codes.

**Implementation**   We implement our method[3] based on PyTorch framework [64]. Following existing GNN-based methods [37], we select a 3-layer encoding function backbone for both datasets. The maximum threshold $\eta$ for the subgraph selection space is set to 3. More experimental details are given in the Appendix B.3.

## 4.2 Performance Comparison in S0 settings

Table 2 shows the overall results across all benchmarks in S0 setting. As can be seen, CSSE-DDI consistently outperforms all baselines on each dataset, demonstrating its effectiveness in searching for data-specific subgraph-based pipelines for DDI prediction task. Among the baselines, subgraph-based methods significantly outperform full-graph-based methods due to their enhanced ability to reason over local subgraph contexts. Within the subgraph-based methods, SEAL, GraIL, SumGNN, and SNRI use a fixed sample strategy to select subgraphs, which may not be optimal for different drug-pair queries.

When it comes to NAS-based method, MR-GNAS and AutoGEL contain well-established search spaces that embrace multi-relational message-passing schema, focusing primarily on automated encoding function design using the one-shot NAS paradigm. While CSSE-DDI adopts a single path supernet training strategy and a message-aware partitioning approach to search for data-adaptive subgraph-based pipelines with stability and robustness, enabling the model to achieve excellent performance across various datasets. Moreover, the consistent performance gains of CSSE-DDI over its two variants validate the importance of jointly customizing subgraph-based pipeline components, i.e., fine-grained subgraphs and data-specific encoding functions, to fit datasets rather than relying on a fixed approach.

Figure 2 shows the learning curves of several competitive methods on both datasets, including CompGCN, KnowDDI and the proposed CSSE-DDI. As can be seen, the searched models not only outperform the baselines but also demonstrate a clear advantage in efficiency, highlighting that enhancing model flexibility and adaptability is essential for improving performance and efficiency.

---

[3]Our code is available at https://github.com/LARS-research/CSSE-DDI.

Table 2: CSSE-DDI achieves the best predictive performance compared to state-of-the-art baselines in DDI prediction. Average and standard deviation of five runs are reported. For these metrics, higher values always indicate better performance.

| Model Type | Dataset | Dataset 1: DrugBank | | | Dataset 2: TWOSIDES | | |
|---|---|---|---|---|---|---|---|
| | Task Type | Multi-class | | | Multi-label | | |
| | Methods | F1 Score | Accuracy | Cohen's $\kappa$ | ROC-AUC | PR-AUC | AP@50 |
| GNN-based | Decagon | $57.35_{\pm0.26}$ | $87.19_{\pm0.28}$ | $86.07_{\pm0.08}$ | $91.72_{\pm0.04}$ | $90.60_{\pm0.12}$ | $82.06_{\pm0.45}$ |
| | GAT | $33.49_{\pm0.36}$ | $77.18_{\pm0.15}$ | $74.20_{\pm0.23}$ | $91.18_{\pm0.14}$ | $89.86_{\pm0.05}$ | $82.80_{\pm0.17}$ |
| | SkipGNN | $59.66_{\pm0.26}$ | $85.83_{\pm0.18}$ | $84.20_{\pm0.16}$ | $92.04_{\pm0.08}$ | $90.90_{\pm0.10}$ | $84.25_{\pm0.25}$ |
| | CompGCN | $71.20_{\pm0.70}$ | $88.30_{\pm0.29}$ | $86.15_{\pm0.35}$ | $93.00_{\pm0.07}$ | $91.26_{\pm0.07}$ | $86.18_{\pm0.10}$ |
| | ACDGNN | $86.24_{\pm0.93}$ | $90.53_{\pm0.38}$ | $87.81_{\pm0.33}$ | $93.69_{\pm0.47}$ | $92.12_{\pm0.21}$ | $87.45_{\pm0.24}$ |
| | TransFOL | $89.97_{\pm1.64}$ | $91.92_{\pm0.89}$ | $90.92_{\pm0.72}$ | $94.16_{\pm0.62}$ | $93.52_{\pm0.53}$ | $88.13_{\pm0.39}$ |
| Subgraph-based | SEAL | $48.82_{\pm0.98}$ | $76.61_{\pm0.26}$ | $71.91_{\pm0.59}$ | $90.74_{\pm0.22}$ | $90.11_{\pm0.17}$ | $84.13_{\pm0.13}$ |
| | GraIL | $73.20_{\pm0.69}$ | $85.40_{\pm0.39}$ | $82.70_{\pm0.47}$ | $92.93_{\pm0.10}$ | $91.69_{\pm0.14}$ | $87.43_{\pm0.09}$ |
| | SumGNN | $78.35_{\pm0.51}$ | $89.05_{\pm0.36}$ | $87.28_{\pm0.08}$ | $92.62_{\pm0.04}$ | $90.80_{\pm0.40}$ | $85.75_{\pm0.10}$ |
| | SNRI | $85.57_{\pm0.32}$ | $90.15_{\pm0.21}$ | $88.94_{\pm0.36}$ | $93.12_{\pm0.18}$ | $92.64_{\pm0.12}$ | $87.53_{\pm0.11}$ |
| | KnowDDI | $90.06_{\pm0.27}$ | $93.15_{\pm0.19}$ | $91.87_{\pm0.21}$ | $95.05_{\pm0.06}$ | $93.75_{\pm0.05}$ | $89.24_{\pm0.06}$ |
| | LaGAT | $81.63_{\pm0.56}$ | $86.21_{\pm0.18}$ | $85.38_{\pm0.23}$ | $89.78_{\pm0.21}$ | $86.33_{\pm0.15}$ | $83.75_{\pm0.36}$ |
| NAS-based | MR-GNAS | $74.24_{\pm0.45}$ | $88.17_{\pm0.24}$ | $87.31_{\pm0.11}$ | $93.85_{\pm0.07}$ | $91.80_{\pm0.03}$ | $87.16_{\pm0.05}$ |
| | AutoGEL | $76.87_{\pm0.63}$ | $89.35_{\pm0.59}$ | $86.14_{\pm0.41}$ | $94.11_{\pm0.32}$ | $92.35_{\pm0.29}$ | $88.13_{\pm0.41}$ |
| | CSSE-DDI-FS | $86.31_{\pm0.36}$ | $91.08_{\pm0.21}$ | $89.17_{\pm0.27}$ | $94.35_{\pm0.07}$ | $93.01_{\pm0.06}$ | $89.08_{\pm0.04}$ |
| | CSSE-DDI-FF | $80.96_{\pm0.65}$ | $90.27_{\pm0.23}$ | $88.69_{\pm0.31}$ | $94.26_{\pm0.08}$ | $92.74_{\pm0.06}$ | $88.91_{\pm0.09}$ |
| | **CSSE-DDI** | $\mathbf{92.08_{\pm0.22}}$ | $\mathbf{95.56_{\pm0.15}}$ | $\mathbf{94.72_{\pm0.26}}$ | $\mathbf{95.47_{\pm0.02}}$ | $\mathbf{94.21_{\pm0.05}}$ | $\mathbf{89.76_{\pm0.05}}$ |

## 4.3 Choices of Search Strategy

To demonstrate the effectiveness of our search strategy, we introduce two variants with different search strategies: (i) **CSSE-DDI w/o MAP**: This variant uses only one trained supernet to serve as a performance evaluator for candidate architectures, instead of generating multiple sub-supernets by Message-Aware Partition (MAP) strategy. (ii) **CSSE-DDI w/o SPOS**: This variant utilizes the message-aware partition strategy to jointly optimize the supernet weights and architectural parameters, without using the Single Path One-Shot (SPOS) strategy [54] .

In Table 3, we compare CSSE-DDI with other variants. As can be seen, the absence of either message-aware partition strategy or sampling-based NAS strategy negatively impacts performance. The performance gains achieved through the message-aware partition strategy arise from using multiple sub-supernets, which provide more accurate performance estima-

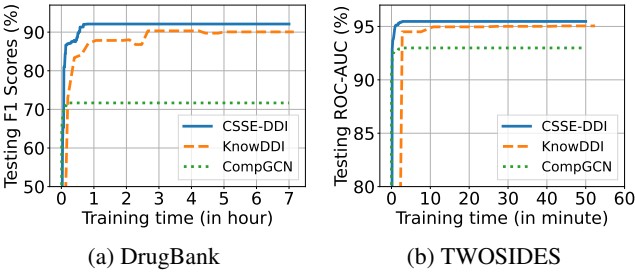

(a) DrugBank  (b) TWOSIDES

Figure 2: Comparison on convergence between the searched architectures by CSSE-DDI and human-designed methods.

Table 3: Performance of CSSE-DDI using different variants of search algorithm.

| Variant | DrugBank | TWOSIDES |
|---|---|---|
| CSSE-DDI w/o MAP | $90.17_{\pm0.29}$ | $95.12_{\pm0.04}$ |
| CSSE-DDI w/o SPOS | $90.97_{\pm0.72}$ | $94.89_{\pm0.13}$ |
| **CSSE-DDI** | $\mathbf{92.08_{\pm0.22}}$ | $\mathbf{95.47_{\pm0.02}}$ |

tions to guide the search process. Regarding the SPOS strategy, it decouples supernet training from architecture search, making it more efficient and robust in practice.

## 4.4 Sensitivity Analysis of the Threshold $\eta$

Here, we analyze the effect of the threshold $\eta$ used in subgraph selection space. Figure 3 shows the impact of varying $\eta$. As can be observed, model performance continues to get better as the threshold $\eta$ grows. When the threshold $\eta = 3$, the model performance nears saturation, as larger thresholds do not lead to further improvements. This is likely because most of the essential information for DDI prediction is contained within the 3-hop ego-subgraphs of target drugs. Intuitively, larger subgraphs may provide additional useful information. However, in practice, due to the inherent biases of the search algorithm, achieving an optimal model may be challenging. When $\eta$ is too large, it may introduce noise and dilute the critical information. A

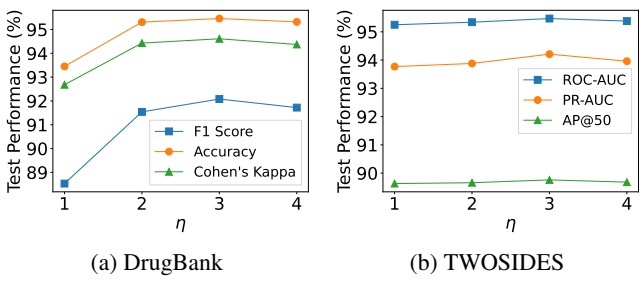

(a) DrugBank          (b) TWOSIDES

Figure 3: Performance given different hyperparameter $\eta$.

similar phenomenon has been found in the existing work SumGNN [12]. Besides, excessively large thresholds $\eta$ will only lead to unnecessary expansion of the search space and higher computational costs.

## 4.5 Performance Comparison in S1 settings

To further validate the effectiveness of our method, we use the S1 setting in the EmerGNN [13] method, to predict drug-drug interactions between emerging drugs and existing drugs. The experimental results are shown in Table 4. A significant performance drop from the transductive setting (S0) to the inductive setting (S1) demonstrates that DDI prediction for new drugs is more challenging. Although Emergnn, which is specifically designed for new drug prediction, achieves optimal performance, CSSE-DDI still demonstrates impressive results, outperforming existing GNN-based and subgraph-based methods. This strong performance is largely due to the robust learning capability of NAS technology in handling unknown data.

Table 4: Experimental results in S1 setting.

| Dataset | Dataset 1: DrugBank | | | Dataset 2: TWOSIDES | | |
|---|---|---|---|---|---|---|
| Task Type | Multi-class | | | Multi-label | | |
| Methods | F1 Score | Accuracy | Cohen's $\kappa$ | ROC-AUC | PR-AUC | Accuracy |
| CompGCN | $30.98_{\pm 3.26}$ | $52.76_{\pm 0.46}$ | $37.87_{\pm 1.28}$ | $84.83_{\pm 1.02}$ | $83.68_{\pm 1.86}$ | $74.64_{\pm 0.79}$ |
| Decagon | $11.39_{\pm 0.79}$ | $32.56_{\pm 0.92}$ | $20.29_{\pm 1.33}$ | $57.49_{\pm 1.75}$ | $59.38_{\pm 1.09}$ | $52.27_{\pm 1.48}$ |
| SumGNN | $26.57_{\pm 1.59}$ | $44.30_{\pm 1.04}$ | $40.24_{\pm 1.26}$ | $80.02_{\pm 2.17}$ | $78.42_{\pm 1.62}$ | $69.81_{\pm 1.77}$ |
| KnowDDI | $31.14_{\pm 1.24}$ | $53.44_{\pm 1.73}$ | $43.93_{\pm 1.17}$ | $84.23_{\pm 2.63}$ | $82.58_{\pm 1.94}$ | $74.72_{\pm 1.51}$ |
| EmerGNN | $\mathbf{58.13}_{\pm \mathbf{1.36}}$ | $\mathbf{69.53}_{\pm \mathbf{1.97}}$ | $\mathbf{62.19}_{\pm \mathbf{1.62}}$ | $\underline{87.42}_{\pm 0.39}$ | $\underline{86.20}_{\pm 0.71}$ | $\underline{79.23}_{\pm 0.54}$ |
| **CSSE-DDI** | $\underline{37.24}_{\pm 1.13}$ | $\underline{58.57}_{\pm 0.85}$ | $\underline{49.97}_{\pm 1.01}$ | $\mathbf{88.33}_{\pm \mathbf{0.52}}$ | $\mathbf{86.47}_{\pm \mathbf{0.27}}$ | $\mathbf{80.01}_{\pm \mathbf{0.39}}$ |

## 4.6 Case Study

### 4.6.1 Fine-grained Subgraph Selection

We visualize exemplar query-specific subgraphs from the DrugBank dataset in Figure 4, highlighting **domain concepts** such as pharmacokinetics, metabolism, and receptor interactions. As shown, CSSE-DDI can identify distinctive subgraphs containing semantic information to support inference for different queries, revealing pharmacokinetic and metabolic relationships.

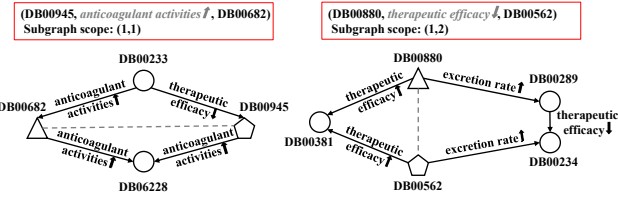

Figure 4: Visualization of the searched subgraphs corresponding to the specific drug pairs.

For example, to predict the interaction between DB00945 (Aspirin) and DB00682 (Warfarin), CSSE-DDI searches out the subgraph scope $(1, 1)$, as depicted on the left part of Figure 4. Firstly, it can be seen from the figure that the therapeutic efficacy of DB00233 (Aminosalicylic acid) can decrease when combined with DB00945 (Aspirin), suggesting similarity between the two drugs [65, 66] Given that DB00233 (Aminosalicylic acid) may increase the anticoagulant activity of DB00682 (Warfarin) and that DB00233 resembles DB00945 (Aspirin), it can be inferred that DB00945 (Aspirin) may similarly increase the anticoagulant activity of DB00682 (Warfarin). This example demonstrates that the identified subgraph contains sufficient semantic information to reason about the interaction between DB00945 (Aspirin) and DB00682 (Warfarin).

### 4.6.2 Data-specific Encoding Function

Furthermore, we visualize the searched structure of encoding functions across all datasets in Figure 5. It is clearly illustrated that different combinations of the designed operations, i.e., data-specific encoding functions, are obtained.

In particular, the searched message-computing functions contain more `CORR` operations in the DrugBank dataset, while more `MULT` functions are searched in the TWOSIDES dataset. The `CORR` function is non-commutative [67], making it suitable for modeling asymmetric interactions (e.g., metabolic-based interactions) present in DrugBank. While `MULT` is suitable for modeling symmetric relations (phenotype-based interactions) due to its exchangeability [68].

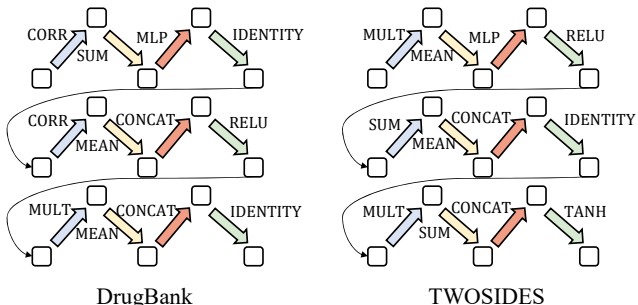

Figure 5: The searched encoding functions on all benchmark datasets.

## 5   Conclusion

We propose a searchable framework, CSSE-DDI, for DDI prediction. Specifically, we design refined search spaces to enable fine-grained subgraph selection and data-specific encoding function optimization. To facilitate efficient search, we introduce a relaxation mechanism to convert the discrete subgraph selection space into a continuous one. Additionally, we employ a subgraph representation approximation strategy to accelerate the search process, addressing the inefficiencies of explicit subgraph sampling. Extensive experiments demonstrate that CSSE-DDI significantly outperforms state-of-the-art methods. Moreover, the search results generated by CSSE-DDI offer interpretability in the context of drug interactions, revealing domain-specific concepts such as pharmacokinetics and metabolism.

## Acknowledgements

We thank the anonymous reviewers for their valuable comments. This work was supported in part by the National Key Research and Development Program of China (Grant No. 2022ZD0160300), in part by the National Science Fund for Distinguished Young Scholars (Grant No. 62025602), in part by the National Natural Science Foundation of China (Grant Nos. U22B2036, 11931015, 62203363, and 92270106), in part by the Technology Innovation Leading Program of Shaanxi (Grant No. 2023GXLH-086), in part by the Beijing Natural Science Foundation (Grant No. 4242039). in part by the Fok Ying-Tong Education Foundation, China (Grant No. 171105), in part by the Fundamental Research Funds for the Central Universities (Grant Nos. G2024WD0151 and D5000240309), and in part by the Tencent Foundation and XPLORER PRIZE.

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

# Appendix

# A  More Method Details

## A.1  Subgraph Encoding Space

An expressive subgraph encoding space can be naturally designed by including human-designed operations, the details of which are given in Table 5.

Table 5: The operations used in our search space.

| Function name | Operations |
|---|---|
| Message Computing Function | SUB, MULT, CORR, ROTATE |
| Aggregation Function | SUM, MAX,MEAN |
| Combination Function | MLP, CONCAT |
| Activation Function | RELU, TANH, IDENTITY |

In particular, given the embedding $\mathbf{h}_u$ of node $u$ and the embedding $\mathbf{h}_r$ of interaction $r$, the message computing function takes the following form: $\text{MES}_{\text{SUB}} = \mathbf{h}_u - \mathbf{h}_r$, $\text{MES}_{\text{MULT}} = \mathbf{h}_u * \mathbf{h}_r$, $\text{MES}_{\text{CORR}} = \mathbf{h}_u \star \mathbf{h}_r$, $\text{MES}_{\text{ROTATE}} = \mathbf{h}_u \circ \mathbf{h}_r$, where $\star$ stands for the circular correlation operation [67], $\circ$ represents the rotation operation [69].

## A.2  Robust Search Algorithm

We adopt the single path one-shot (SPOS) training strategy to solve the customized search problem, which decouple supernet training and architecture searching. In particular, definition 1 can be transformed into a two-step optimaztion [54]:

$$\arg\max_{\boldsymbol{\alpha}\in\mathcal{A},\mathcal{G}_{u,v}\in\mathcal{S}_{u,v}} \sum_{(u,r,v)\in\mathcal{D}_{\text{val}}} \mathcal{M}(\mathbf{W}^*;\mathcal{G}_{u,v};\boldsymbol{\alpha}), \tag{8}$$

$$\mathbf{W}^* = \arg\min_{\mathbf{W}} \mathbb{E}_{\boldsymbol{\alpha}\in\mathcal{A}} \sum_{(u,r,v)\in\mathcal{D}_{\text{tra}}} \mathcal{L}(\mathbf{W};\mathcal{G}_{u,v};\boldsymbol{\alpha}), \tag{9}$$

where $\mathbf{W}$ denotes the shared learnable weights in the supernet with its optimal value $\mathbf{W}^*$ for all the architectures in the overall search space.

Eq. (9),(8) represent the supernet training and architecture searching phase, respectively. In the following, we will describe the detailed process of the two phases.

### A.2.1  Supernet Training

In supernet training phase, a sub-model $\boldsymbol{\alpha}$ is sampled according to the discrete distribution $\pi(\mathcal{A})$. Thus, Eq. (9) can be formulated as

$$\mathbf{W}^* = \arg\min_{\mathbf{W}} \mathbb{E}_{\boldsymbol{\alpha}\sim\pi(\mathcal{A})} \sum_{(u,r,v)\in\mathcal{D}_{\text{tra}}} \mathcal{L}(\mathbf{W};\mathcal{G}_{u,v};\boldsymbol{\alpha}), \tag{10}$$

where the discrete distribution $\pi(\mathcal{A})$ is set to uniform distribution.

First, we need to perform single path sampling to train the supernet until it converges. In the next step, we need to partition the supernet into sub-supernets. which is a key step aiming to isolate operations that are coupled with each other. This allows the supernet to be trained and converge more stably.

In our supernet, we use a message-aware partitions strategy due to the fact that the degree of dissimilarity between the operations in the message computing function MES is much higher compared with others. These operations focus on capture different semantic types of interactions, which has been discussed in existing works [70, 69, 58]. Therefore, we partition four operations of the message computing function of the first layer of the supernet, to improve the accuracy of the performance estimation.

After partitioning operation, we initialize four sub-supernets with weights transferred from the original supernet. Next, we train these sub-supernets to convergence by sampling single path. Here, the supernet training phase is all done.

### A.2.2 Architecture Searching

After completing sub-supernet training phase, we have obtained well-trained supernet weights. In the searching phase, Eq. (8) can be transformed as

$$\arg\max_{\mathcal{G}_{u,v} \in \mathcal{S}_{u,v}} \sum_{(u,r,v) \in \mathcal{D}_{\text{val}}} \mathcal{M}(\mathbf{W}^*; \mathcal{G}_{u,v}; \boldsymbol{\alpha}), \tag{11}$$

$$\text{s.t. } \arg\max_{\boldsymbol{\alpha} \in \mathcal{A}} \sum_{(u,r,v) \in \mathcal{D}_{\text{val}}} \mathcal{M}(\mathbf{W}^*; \mathcal{G}_{u,v}; \boldsymbol{\alpha}), \tag{12}$$

For suugraph encoding function searching in Eq. (12), following [71], we adopt stochastic relaxation on $\boldsymbol{\alpha}$ and natural policy gradient strategy [72] to obtain the optimal subgraph encoding function $\boldsymbol{\alpha}^*$. For subgraph selection in Eq. (11), we obtain the optimal subgraph $\mathcal{G}_{u,v}^*$ by preserving the subgraph with the largest probability $p_{u,v}^{i,j}$, i.e.,

$$\mathbf{z}_{u,v}^{i,j} = f(\mathcal{G}_{u,v}^{i,j}), \tag{13}$$

$$\beta_{u,v}^{i,j} = g(\mathbf{z}_{u,v}^{i,j}), \tag{14}$$

$$p_{u,v}^{i,j} = \frac{\exp(\log(\beta_{u,v}^{i,j} + \mathbf{G}_{i,j})/\tau)}{\sum_{i',j'=1}^{\eta} \exp(\log(\beta_{u,v}^{i',j'} + \mathbf{G}_{i',j'})/\tau)}, \tag{15}$$

$$\mathcal{G}_{u,v}^* = \arg\max_{\mathcal{G}_{u,v}^{i,j}} p_{u,v}^{i,j} (\mathcal{G}_{u,v}^{i,j} \in S_{u,v}). \tag{16}$$

## B    More Experiment Setting

### B.1    Datasets

Experiments are performed on two public benchmark DDI datasets: DrugBank and TWOSIDES.

**DrugBank**    DrugBank dataset contains 1,710 drugs and drug pairs, which are related to 86 types of pharmacological interactions between drugs, such as increase of anticoagulant activity, decrease of excretion rate and etc.

**TWOSIDES**    TWOSIDES dataset contains 604 drugs and drug pairs with 200 drug side effects as interaction labels. For each edge, it may be associated with multiple interactions.

The detailed descriptions for datasets are presented in Table 6 and Table 7.

Table 6: The statistics of the datasets.

| Dataset | #nodes | #edges | #interaction types |
|---------|--------|--------|--------------------|
| DrugBank | 1,710 | 134641 | 86 |
| TWOSIDES | 604 | 57778 | 200 |

Table 7: Diverse semantic properties in drug-drug interactions.

| Dataset | Interaction Type | Examples | Semantic Property |
|---------|------------------|----------|-------------------|
| DrugBank | Metabolic levels-based | #Drug1 may decrease the excretion rate of #Drug2 | asymmetry $(r(x,y) \not\Rightarrow r(y,x))$ |
| TWOSIDES | Phenotype-based | Combination of #Drug 1 and #Drug 2 may cause headaches | symmetry $(r(x,y) \Rightarrow r(y,x))$ |

### B.2    Evaluation Metric

We follow [12] to evaluate our method. Specifically, in terms of the multi-class prediction on DrugBank, we followc[12] and evaluate the performance by three metrics: (i) Macro F1 score (Macro F1) is computed by taking the arithmetic mean (aka unweighted mean) of all the per-class F1 scores. (ii) Accuracy (ACC) is calculated by dividing the number of correct predictions by the total prediction

number. (iii) Coken's Kappa (Cohen's $\kappa$) measures inter-rater reliability. As to the multi-label prediction on TWOSIDES, we consider the following measure and use the average performance over all interaction types: (i) ROC-AUC (AUROC) stands for "Area Under the Curve (AUC)" of the "Receiver Operating Characteristic (ROC)" curve. (ii) PR-AUC (AUPRC) is the average area under precision-recall curve. (iii) AP@50 is the average precision at 50.

### B.3 Implementation and Hyperparameters

All the experiments are implemented in Python with the PyTorch framework [64] and run on a server machine with single NVIDIA RTX 3090 GPU with 24GB memory and 64GB of RAM. Our code is added in the supplementary material.

For CSSE-DDI, we set the epoch to 400 for training supernet and set the epoch to 400 for training sub-supernets. We set the the temperature parameter as 0.05. Repeat 5 times with different seeds, we can get 5 candidates. The searched candidates are finetuned individually with the hyper-parameters. In the stage of fine-tuning, we use the ReduceLROnPlateau scheduler to adjust the learning rate dynamically. Each candidate has 10 hyper steps. In each hyper step, a set of hyperparameter will be sampled from Table 8.

Table 8: Hyperparameters we used during the fine-tuning stage.

| Hyperparameter | Value range |
|---|---|
| Learning rate | $[10^{-3.1}, 10^{-2.9}]$ |
| Weight decay | $[10^{-5}, 10^{-3}]$ |

## C  More Experimental Results

### C.1  Subgraph Scope Distribution Analysis

We visualize the learned distributions of subgraph scope on all datasets by using CSSE-DDI in Figure 6. By comparing the distributions across different benchmarks, we have the following observation: CSSE-DDI can effectively learn different subgraph scope distributions for various datasets. By identifying specific subgraph scopes for different queries, CSSE-DDI is able to precisely control the extent of information propagation required for reasoning about the interactions of different drug pairs. In addition, our method can skip some subgraph scopes if they are not optimal for any queries. For example, no queries are assigned to the propagation scope $(3, 3)$ on TWOSIDES dataset. It is worth mentioning that our searched subgraph scopes are consistent with the sensitivity analysis results for the hop of subgraph in SumGNN [12], which further validates the effectiveness of our approach.

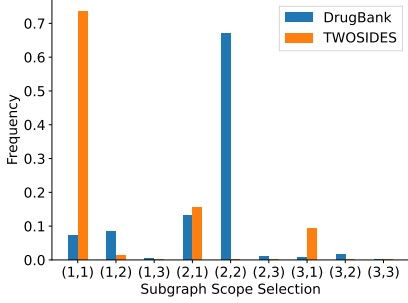

Figure 6: Distribution of the searched subgraph scopes by CSSE-DDI on all benchmark datasets.


## D.1 Limitations

There are three limitations for CSSE-DDI. (1) CSSE-DDI is focused on method design rather than system design. In the future, we will co-design the algorithm and the system to further improve the efficiency. (2) At present, CSSE-DDI only search for data-specific components of subgraph-based pipeline, while hyper-parameters are also important for DDI prediction. A promising direction is to explore how to efficiently search network architectures and hyper-parameters simultaneously.

