# OpenReview forum: "Customized Subgraph Selection and Encoding for Drug-drug Interaction Prediction"
_NeurIPS.cc/2024/Conference — NeurIPS 2024 poster_

### Official Review · Reviewer_KE8i · 2024-06-22

**Soundness:** 4
**Presentation:** 4
**Contribution:** 4
**Rating:** 8
**Confidence:** 5

**Summary:**

The paper proposes a novel subgraph-based approach for predicting drug-drug interactions (DDIs) that harnesses neural architecture search (NAS) to customize subgraph selection and encoding process. The authors first introduce refined search spaces to realize fine-grained subgraph selection and expressive encoding function searching. Then, based on the well-defined bi-level search problem, the subgraph space relaxation mechanism and the representation approximation strategy are proposed, enabling differentiable searching efficiently. Extensive experiments show that CSSE-DDI extensively outperforms the state-of-the-art approaches.

**Strengths:**

1.The problem and the method are well-motivated and formulated, with extensive experimental support. The results are solid.
2.The idea of customizing subgraph selection and encoding is a conceptually strong and justified innovation, which can distinguish this paper from other approaches, as shown in Table 1.
3.The manuscript is well-structured and clearly written, facilitating comprehension.
4.CSSE-DDI demonstrates superior performance on two recognized benchmarks, indicating its practical effectiveness. The case study shows interpretability in the context of drug interactions, which is a very important aspect in such an application paper.

**Weaknesses:**

1. For the supernet training phase, in addition to the algorithm process, the authors are encouraged to provide an illustration to help the reader understand the corresponding steps more clearly.
2. A typo error in line 220: “Subgraph Repersentation“

**Questions:**

Given the method itself is general, is there any reason why the authors would like to specifically focus on DDI prediction?

**Limitations:**

The authors have adequately addressed the limitations.

---

> ### Author Rebuttal · Authors · 2024-08-07
>
> Thank you for your time and efforts in reviewing our paper. Please find our responses below to your concerns.
>
> > W1.	For the supernet training phase, in addition to the algorithm process, the authors are encouraged to provide an illustration to help the reader understand the corresponding steps more clearly.
>
> **R1**. Thanks to your suggestion, we will add a graphical illustration of the search process in the revised version to show the supernet training process more clearly.
>
> > W2.	A typo error in line 220: “Subgraph Repersentation“
>
> **R2**. Thank you for your careful review! We have thoroughly checked the entire text and corrected all spelling and formatting issues.
>
> > Q1.	Given the method itself is general, is there any reason why the authors would like to specifically focus on DDI prediction?
>
> **R3**. DDI is a critical task in pharmacology and healthcare. Compared with the generalized multi-relational graph link prediction, DDI prediction has more explicit application scenarios and practical needs. Therefore, we design CSSE-DDI based on the DDI prediction task and dataset. However, our method can also be used for generalized multi-relational graph link prediction, i.e., knowledge graph relation prediction, after the modification of search space, And we will include some discussions about knowledge graph relation prediction in the revised version.

---

### Official Review · Reviewer_zvSg · 2024-06-25

**Soundness:** 3
**Presentation:** 3
**Contribution:** 3
**Rating:** 3
**Confidence:** 4

**Summary:**

The article presents a novel method for predicting drug-drug interactions (DDIs) by using a customized subgraph selection and encoding process. The authors propose a framework called Customized Subgraph Selection and Encoding for Drug-Drug Interaction prediction (CSSE-DDI), which leverages neural architecture search (NAS) to tailor the subgraph selection and encoding components for different datasets.Extensive experiments demonstrating superior performance of the CSSE-DDI framework compared to hand-designed methods.

**Strengths:**

1.The use of NAS to customize subgraph selection and encoding for DDI prediction is a novel approach that addresses the limitations of fixed, hand-designed methods.

2. The creation of extensive subgraph selection and encoding spaces allows for more accurate and context-specific predictions.

3. The framework's ability to adapt to different datasets and customize subgraph components based on data-specific characteristics is a significant strength.

**Weaknesses:**

1.The author applies NAS to the DDI task, but it does not exhibit the task-specific customization expected for DDI. For instance, the primary focus of the DDI task should be the interactions between drugs. However, the author raises the following issues: "The relaxation strategy is a prerequisite for differentiable NAS methods. This is because the subgraphs in the selection space comprise different nodes and edges, making it challenging to design a relaxation function that unifies subgraphs of varying sizes. Additionally, to search within the subgraph selection space, we must first obtain all subgraphs within this space, but the task of sampling such a large number of subgraphs is computationally infeasible." These issues are also present in subgraph extraction tasks on a single graph.

2.To address the problem of subgraph sampling, the author employs an implicit encoding sampling method. However, its effectiveness in terms of time efficiency and accuracy lacks sufficient evidence.

3.In the DDI task, the subgraph sampling space grows exponentially. The author's method for addressing the excessively large sampling space does not offer any advantages over the methods used to address the large sampling space in a single graph.

4.The author does not provide a comparison of the time overhead between their NAS method and the baseline methods. Given that NAS is likely to have significantly high time overhead, this comparison is crucial for evaluating model performance.

5.The author lacks comparisons with the latest baselines to highlight the performance advantages of their model. For instance, only one of the baselines chosen by the author is from 2023.

**Questions:**

If the author solves the above weaknesses, I am willing to change my score.

In addition, the author's method solves the subgraph sampling problem. Whether the model can perform visual analysis on the sampled subgraphs is critical for the interpretability of the model.

If possible, I hope the author can provide more baseline comparisons and include a greater variety of experimental settings (such as inductive settings, which are common in many DDI tasks), as well as experiments on more datasets.

---

> ### Author Rebuttal · Authors · 2024-08-07
>
> Thank you for your time and efforts in reviewing our paper. Please find our responses below to your concerns.
>
> > W1. About task-specific customization expected for DDI.
>
> **R1**. Please refer to the general response **GR1**.
>
> > W2. Time efficiency and accuracy about implicit encoding sampling method.
>
> **R2**. The time overhead of using explicit subgraphs is huge. When we use explicit subgraphs, we need to sample all candidate subgraphs of a query. For $\eta=3$ on Drugbank, the sampling time is 15 hours. In the process of searching, the memory overhead will be multiple times that of the subgraph-based method.
>
> However we can prove the effectiveness of our method in the following way: We collect the corresponding subgraph sizes of different queries obtained from the search process, explicitly sample the subgraphs of the corresponding sizes and use the searched message functions to encode the subgraphs and predict the interactions between drug pairs， called CSSE-DDI(explicit). Table 4 shows the experimental results. It can be seen that the prediction performance of explicit sampling and implicit sampling is similar. However, in terms of single running time, the explicit sampling method needs to perform a separate message passing on each subgraph, whereas the implicit sampling method is more efficient as it can quickly obtain subgraph representations using the subgraph representation approximation strategy.
>
> |Table 4|Drugbank|||
> |-------|-------|-------|-------|
> | |F1|ACC|Running time(minutes)|
> |CSSE-DDI(explicit)|91.63±0.29|94.23±0.54|201|
> |**CSSE-DDI**|92.08±0.22|95.56±0.15|32|
>
> > W3. About the advantage of the proposed subgraph sampling space.
>
> **R3**. Please refer to the general response **GR1**.
>
> > W4. About time overhead between their NAS method and the baseline methods.
>
> **R4**. Please refer to the general response **GR2**.
>
> > W5. Lack comparisons with the latest baselines.
>
> **R5**. Thanks for the suggestion. We add the results of comparisons with multiple latest baselines, as shown in Table 6, including LaGAT[1] published in 2022, ACDGNN[2] published in 2023, and TransFOL[3] published in 2024. All codes are publicly available.
>
> |Table 6|Drugbank||TWOSIDES||
> |-------|-------|-------|-------|-------|
> | |F1|ACC|ROC-AUC|PR-AUC|
> |LaGAT (2022)|81.63±0.56|86.21±0.18|89.78±0.21|86.33±0.15|
> |ACDGNN (2023)|86.24±0.93|90.53±0.38|93.69±0.47|92.12±0.21|
> |TransFOL (2024)|89.97±1.64|91.92±0.89|94.16±0.62|93.52±0.53|
> |**CSSE-DDI**|92.08±0.22|95.56±0.15|95.47±0.02|94.21±0.05|
>
> As can be seen, CSSE-DDI is still competitive against the latest baselines. For LaGAT[1], it still extracts the same range of subgraphs for different queries, and in addition R-GAT is still a suboptimal solution for handling diverse DDI interaction data. For ACDGNN, its overall framework is still based on a complete bio-heterogeneous network for reasoning, and its expressive power is still limited compared with the subgraph-based approach.For TransFOL, which utilizes cross-transformers and graph convolutional networks to deal with interactions in DDI datasets, it mainly focuses on complex logical query tasks in DDI data, and does not have any advantage in single drug-drug interaction prediction tasks.
>
> > Q1. About visual analysis.
>
> **R6**. We visualize some of the sampled subgraphs in Section 4.5.1 and make some case studies to demonstrate the effectiveness and interpretability of our method. In the revised version, we will add more visual analysis.
>
> > Q2. Provide more baseline comparisons, inductive setings, as well as experiments on more datasets.
>
> **R7**. For the experimental results of the latest baselines, please refer to **R5**.
>
> In terms of inductive settings. we use the inductive setting and datasets in EmerGNN[4] method, to predict drug-drug interactions between emerging drugs and existing drugs. The experimental results are shown in Table 7. Emergnn, which is specifically designed for new drug prediction, achieves optimal performance. However, CSSE-DDI still achieves impressive performance, which is mainly due to the robust learning ability of NAS technology for unknown data.
>
> |Table 7|Drugbank||TWOSIDES||
> |-------|-------|-------|-------|-------|
> | |F1|ACC|ROC-AUC|PR-AUC|
> |CompGCN|30.98±3.26|52.76±0.46|84.83±1.02|83.68±1.86|
> |SumGNN|26.57±1.59|44.30±1.04|80.02±2.17|78.42±1.62|
> |KnowDDI|31.14±1.24|53.44±1.73|84.23±2.63|82.58±1.94|
> |EmerGNN|58.13±1.36|69.53±1.97|87.42±0.39|86.20±0.71|
> |**CSSE-DDI**|37.24±1.13|58.57±0.85|88.33±0.52|86.47±0.27|
>
> For more datasets, most of the research work on DDI prediction only uses the DrugBank and TWOSIDES datasets, but we have been continually looking to see if there are other available datasets to validate the effectiveness of our approach.
>
> [1] LaGAT: link-aware graph attention network for drug–drug interaction prediction, Bioinformatics' 22.
>
> [2] Attention-based cross domain graph neural network for prediction of drug–drug interactions, Briefings in Bioinformatics' 23.
>
> [3] TransFOL: A Logical Query Model for Complex Relational Reasoning in Drug-Drug Interaction, IEEE Journal of Biomedical and Health Informatics' 24.
>
> [4] Emerging Drug Interaction Prediction Enabled by Flow-based Graph Neural Network with Biomedical Network. Nature Computational Science' 23.

---

> > ### Comment · Reviewer_zvSg · 2024-08-14
> >
> > Your reply has solved most of my questions, but I still have some doubts about the innovation of the method: the customization of the DDI prediction task is limited to building a search space that adapts to the DDI dataset. I think the innovation is limited, and it does not combine the DDI task to innovate the method and theory. If there is, I hope the author can further clarify it. Otherwise, I think this paper simply transfers the method applied on a single molecule to the DDI task.

---

> > > ### Author Response · Authors · 2024-08-14
> > >
> > > Thank you for your patience and positive feedback in helping us improve our work. We would like to kindly argue that CSSE-DDI introduces the crucial task-specific customization for DDI prediction. A comprehensive and thorough analysis demonstrates our insight of the core issues in the drug-drug interaction (DDI) prediction task, which has also prompted us to utilize NAS technology for precise prediction of drug-drug interactions. For NAS-based methods, the innovation typically manifests in the design of a search space informed by domain insights, coupled with search strategies capable of efficiently exploring the proposed space. Therefore, we wish to emphasize that our manuscript presents clear motivations, novel, and effective designs specifically tailored for the drug-drug interaction (DDI) prediction problem.

---

### Official Review · Reviewer_2Vrd · 2024-07-09

**Soundness:** 2
**Presentation:** 2
**Contribution:** 2
**Rating:** 4
**Confidence:** 4

**Summary:**

This work introduces CSSE-DDI, a searchable framework for DDI prediction, which refines search spaces for fine-grained subgraph selection and data-specific encoding. To improve search efficiency, CSSE-DDI employs a relaxation mechanism to continuousize the discrete subgraph selection space and use subgraph representation approximation to accelerate the search process. Extensive experiments demonstrate that CSSE-DDI significantly outperforms state-of-the-art methods, and the results are interpretable, revealing domain concepts like pharmacokinetics and metabolism.

**Strengths:**

it introduces comprehensive subgraph selection and encoding spaces to cover the diverse contexts of drug interactions for DDI prediction. Faced with overwhelming sampling overhead, this work designs an effective relaxation mechanism to efficiently explore optimal subgraph configurations using an approximation strategy, enabling a robust search algorithm to explore the search space efficiently.

**Weaknesses:**

Examples of symmetric semantic patterns(headache, pain in throat) are not very convincing.

**Questions:**

CSSE-DDI-FF and KnowDDI both are fine-grained Subgraph Selection. What is the reason for the performance difference？
Whether CSSE-DDI can be used to predict ddi for new drugs?

---

> ### Author Rebuttal · Authors · 2024-08-07
>
> Thank you for your time and efforts in reviewing our paper. Please find our responses below to your concerns.
>
> > W1. Examples of symmetric semantic patterns(headache, pain in throat) are not very convincing.(sematic property)
>
> **R1**. In the field of multi-relational graphs, the modeling of different semantic types of interaction relationships has been a research hotspot. Many works[1-4] have tried to propose diverse hand-designed interaction functions to model different properties of semantic attributes. In the DDI dataset, the semantic nature of drug interactions is diverse, and our example for asymmetric type and symmetric type is to validate the semantic diversity in different datasets. Such diversity is what motivates us to use NAS to search for adaptive models.
>
> > Q1.1 CSSE-DDI-FF and KnowDDI both are fine-grained Subgraph Selection. What is the reason for the performance difference？
>
> **R2.1**. It should be noted here that the query subgraph of KnowDDI, which is extracted by combining external biological knowledge graphs (HetioNet), contains richer external information, such as genes, diseases, proteins, etc. We label KnowDDI as a fine-grained subgraph selection method, which means that it utilizes a graph structure learning module to realize a fine-grained information supplementation and pruning of the extracted subgraphs. Whereas, although our variant (CSSE-DDI-FF) carries out fine-grained subgraph selection, the variant only employs a basic GNN backbone (CompGCN) to encode subgraphs without external knowledge, which makes it difficult to effectively capture the semantic information in the subgraphs, and thus lead to some performance differences.
>
> And CSSE-DDI still outperforms KnowDDI without relying on external knowledge by fine-grained subgraph selection and data-specific encoding function, which directly demonstrates the effectiveness of our method in capturing complex interactions in DDI datasets.
>
> > Q1.2 Whether CSSE-DDI can be used to predict DDI for new drugs?
>
> **R2.2**. As the reviewer pointed out, there is a strong need to predict DDI related to new drugs in real-world scenarios. Therefore, to further validate the effectiveness of our method, we use the inductive setting and datasets in the EmerGNN[5] method, to predict drug-drug interactions between emerging drugs and existing drugs. The experimental results are shown in Table 3. It can be seen that there is a significant performance drop from the transductive setting to the inductive setting, which shows that the DDI prediction for new drugs is more difficult. Emergnn, which is specifically designed for new drug prediction, achieves optimal performance. However, CSSE-DDI still achieves impressive performance, which is mainly due to the robust learning ability of NAS technology for unknown data.
>
> |Table 3|Drugbank||TWOSIDES||
> |-------|-------|-------|-------|-------|
> | |F1|ACC|ROC-AUC|PR-AUC|
> |CompGCN|30.98±3.26|52.76±0.46|84.83±1.02|83.68±1.86|
> |Decagon|11.39±0.79|32.56±0.92|57.49±1.75|59.38±1.09|
> |SumGNN|26.57±1.59|44.30±1.04|80.02±2.17|78.42±1.62|
> |KnowDDI|31.14±1.24|53.44±1.73|84.23±2.63|82.58±1.94|
> |EmerGNN|58.13±1.36|69.53±1.97|87.42±0.39|86.20±0.71|
> |**CSSE-DDI**|37.24±1.13|58.57±0.85|88.33±0.52|86.47±0.27|
>
> **References**
>
> [1] Translating Embeddings for Modeling Multi-relational Data, NeurIPS' 13.
>
> [2] Embedding Entities and Relations for Learning and Inference in Knowledge Bases, ICLR' 15.
>
> [3] Holographic Embeddings of Knowledge Graphs, AAAI' 16.
>
> [4] RotatE: Knowledge Graph Embedding by Relational Rotation in Complex Space, ICLR' 19.
>
> [5] Emerging Drug Interaction Prediction Enabled by Flow-based Graph Neural Network with Biomedical Network. Nature Computational Science' 23.

---

### Official Review · Reviewer_Qw7Z · 2024-07-12

**Soundness:** 3
**Presentation:** 3
**Contribution:** 3
**Rating:** 5
**Confidence:** 3

**Summary:**

This paper addresses the challenge of predicting drug-drug interactions (DDIs), crucial for medical practice and drug development, using subgraph-based methods. It highlights the importance of customizing subgraph selection and encoding but notes the high cost of manual adjustments. Inspired by neural architecture search (NAS), the authors propose a method to search for data-specific components in the subgraph-based pipeline. They introduce extensive subgraph selection and encoding spaces and design a relaxation mechanism to efficiently explore optimal configurations. Extensive experiments demonstrate the method's effectiveness and adaptability.

**Strengths:**

1. Extensive experiments demonstrate the method's effectiveness. Compared to existing hand-designed methods, the CSSE-DDI framework shows superior performance, enhancing the proposed method's validity.
2. The writing quality is good.
3. Using NAS for precise prediction of drug-drug interactions is novel.

**Weaknesses:**

1. The motivation for using NAS to search components is not clear.
2. Despite the designed efficiency mechanisms, the inherent overhead of neural architecture search remains significant, especially for large-scale DDI prediction tasks.
3. While the paper compares the proposed method with existing ones, a more detailed comparative analysis, including discussions on computational costs and efficiency metrics, would provide a clearer picture of the trade-offs involved.

**Questions:**

1. Why is the performance of GAT in DrugBank so poor? In [1], it seems to work well.
2. Why does the NAS method proposed by the authors appear to perform much better than other NAS methods in the DrugBank dataset? What is the core reason for this?

[1] Hong Y, Luo P, Jin S, et al. LaGAT: link-aware graph attention network for drug–drug interaction prediction[J]. Bioinformatics, 2022, 38(24): 5406-5412.

**Limitations:**

See weakness.

---

> ### Author Rebuttal · Authors · 2024-08-07
>
> Thank you for your time and efforts in reviewing our paper. Please find our responses below to your concerns.
>
> > W1. The motivation for using NAS to search components is not clear.
>
> **R1**. The use of NAS technology to customize components stems from our analysis and understanding of the DDI prediction problem and datasets in the following two ways:
>
> 1.	Different DDI datasets have different semantic natures of interactions. (asymmetric patterns in DrugBank and symmetric in TWOSIDES)
> 2.	Reasoning that different drug-pair queries should require different ranges of subgraph information, rather than a fixed range. (Our empirical analysis in the supplementary PDF(Figure A1))
>
> Based on the above analysis, we believe that in the face of diverse data properties, the use of customized search techniques, i.e., NAS, can help improve the performance of DDI prediction task and the interpretability of the results, as evidenced by our extensive experiments.
>
> > W2. Despite the designed efficiency mechanisms, the inherent overhead of neural architecture search remains significant, especially for large-scale DDI prediction tasks.
>
> **R2**. We would like to kindly argue that our design for the search algorithm with the implicit subgraph sampling strategy is precisely to cope with the large-scale graph data. Tab. I shows the running overhead of our method and other subgraph methods on the drugbank dataset, for the subgraph methods, before the "+" sign represents the explicit subgraph sampling overhead, and for our method, before the "+" sign represents the searching overhead, and after the "+" sign represents the model's single running time.
>
> As can be seen, CSSE-DDI and the subgraph approach are comparable or even superior in terms of time overhead, which stems from the fact that our implicit subgraph sampling strategy saves a lot of time in the subgraph sampling and representation phases. It is conceivable that if the size of the dataset is further increased, the number of subgraph sampling will be further increased and the advantage of our implicit sampling strategy will be even more significant.
>
> |Table 1|Running time(minutes)|
> |-------|-------|
> |SumGNN|342+361|
> |KnowDDI|379+393|
> |**CSSE-DDI**|444+32|
>
> > W3. While the paper compares the proposed method with existing ones, a more detailed comparative analysis, including discussions on computational costs and efficiency metrics, would provide a clearer picture of the trade-offs involved.
>
> **R3**.  For the discussion of the time overhead, please refer to **R2**.
> Overall, compared with subgraph methods, CSSE-DDI has lower time overheads, which is the advantage brought by our proposed implicit subgraph sampling strategy. Compared with GNN-based methods, CSSE-DDI, by analyzing the DDI prediction problem and datasets, designs a search space that can be adapted to different data by analyzing the DDI prediction problem. By efficiently searching for the components, it substantially improves the prediction performance under tolerable time overhead compared with the traditional GNN-based methods. In addition, CSSE-DDI can design robust and customized model structures to cope with unknown datasets, which is exactly the advantage of utilizing NAS technology to solve the DDI prediction problem.
>
> > Q1.1. Why is the performance of GAT in DrugBank so poor?
>
> **R4.1**. A large number of literature[1-3] points to the fact that although GAT[4] considers the attention on different edges, it fails to consider more generalized and more complex multi-relational graph data, which contains diverse semantic information, as its performance is also not good. Therefore, the direct application of GAT on DDI graphs results in suboptimal solutions.
>
> > Q1.2. In [3], it seems to work well.
>
> **R4.2**. Compared with GAT, LaGAT extends the graph encoding module of SumGNN, and utilizes multi-relational GAT to take into account the diverse types of relations and the attention weights of different edges in the DDI data. We re-run its code in the DrugBank and  TWOSIDES data, and the comparison results are shown in Table 2. As shown below, our approach still achieves the best performance by customizing subgraph selection and encoding process. For the performance reported by LaGAT, we were unable to reproduce it. The official repository does not release the corresponding dataset, although it claims it uses the same DrugBank dataset as SumGNN.
>
> |Table 2|Drugbank||TWOSIDES||
> |-------|-------|-------|-------|-------|
> | |F1|ACC|ROC-AUC|PR-AUC|
> |LaGAT|81.63±0.56|86.21±0.18|89.78±0.21|86.33±0.15|
> |**CSSE-DDI**|92.08±0.22|95.56±0.15|95.47±0.02|94.21±0.05|
>
> > Q2. Why does the NAS method proposed by the authors appear to perform much better than other NAS methods in the DrugBank dataset? What is the core reason for this?
>
> **R5**. Compared with other NAS methods, our method has advantages in both search space and search algorithm.
>
> - **Search space**: we design subgraph selection spaces and subgraph encoding spaces adapted to the DDI dataset (Sec 3.2), whereas other methods are only limited to generalized multi-relational graphs and do not have task-specific customization for the DDI prediction task.
>
> - **Search algorithm**: we design a message-aware partition supernet training strategy for the operation coupling problem(Sec 3.3.4), which improves the consistency and accuracy of supernet, enabling the search algorithm more stable and robust, whereas the other methods only use the traditional one-shot search algorithm, which leads to sub-optimal performance.
>
> **References**
>
> [1] SumGNN: Multi-typed Drug Interaction Prediction via Efficient Knowledge Graph Summarization, Bioinformatics' 21.
>
> [2] r-GAT: Relational Graph Attention Network for Multi-Relational Graphs. Arxiv' 21.
>
> [3] LaGAT: link-aware graph attention network for drug–drug interaction prediction, Bioinformatics' 22.
>
> [4] Graph Attention Networks, ICLR' 18.

---

> > ### Comment · Reviewer_Qw7Z · 2024-08-13
> >
> > Thank you for the author's response. I have carefully read the rebuttal, and the analysis of the motivation and the efficiency experiments have largely resolved my doubts. However, I still have some minor questions regarding some experimental results. GAT performed quite well in the literature [3], so why did it perform only moderately in the author's experimental results? I will increase my rate to 5.

---

> > > ### Author Response · Authors · 2024-08-14
> > >
> > > Thank you for your patience and suggestions in helping us improve our work. For all baseline results (including GAT [4]), we reran the official code and fine-tuned the hyperparameters to ensure consistency with the conclusions of existing literature [1]. Regarding the GAT results in [3], based on our experience, the results appear to be on the higher side. We will reach out to the authors of [3] to obtain more details about the results.

---

### Author Rebuttal · Authors · 2024-08-07

Dear reviewers,

Thank you for your time and comments in reviewing our paper. To summarize, all reviewers agree that **the use of NAS to customize subgraph selection and encoding for DDI prediction is a novel approach** (Qw7Z, zvSg, KE8i, 2Vrd). **The manuscript is well-structured and clearly written, facilitating comprehension** (Qw7Z, KE8i). **The problem and the method are well-motivated and formulated, with extensive experimental support** (zvSg, KE8i). The proposed method, CSSE-DDI, **addresses the DDI prediction problem effectively** (Qw7Z, zvSg, KE8i). The case study **shows interpretability in the context of drug interactions, which is a very important aspect in such an application paper** (KE8i).

We believe all of the reviewers’ concerns can be addressed. In the following, we respond to the main concerns and suggestions raised in the review:

### GR1. Task-specific customization for DDI (zvSg, Qw7Z)

Our customization for the DDI prediction task is the search space adapted to the DDI datasets:

- **Search Space**: We design subgraph selection spaces **(the dense property of DDI dataset)** and subgraph encoding spaces **(the diverse semantic nature of DDI dataset, i.e., asymmetric patterns in DrugBank and symmetric in TWOSIDES)** adapted to the DDI dataset (Sec 3.2).

### GR2. The inherent overhead of neural architecture search (Qw7Z, zvSg)

We would like to kindly argue that **our design for the search algorithm with the implicit subgraph sampling strategy is efficient**. Table G1 shows the running overhead of our method and other subgraph methods on the DrugBank dataset, for the subgraph methods, before the "+" sign represents the explicit subgraph sampling overhead, and for our method, before the "+" sign represents the searching overhead, and after the "+" sign represents the model's single running time.

|Table G1|Running time(minutes)|
|-------|-------|
|SumGNN|342+361|
|KnowDDI|379+393|
|**CSSE-DDI**|444+32|


In general, **CSSE-DDI and the subgraph approach are comparable or even superior in terms of time overhead**, which stems from the fact that our implicit subgraph sampling strategy saves a lot of time in the subgraph sampling and representation phases.

### GR3. Inductive setting (Qw7Z,2Vrd)

There is a strong need to predict DDI related to new drugs in real-world scenarios. Therefore, to further validate the effectiveness of our method, we use the inductive setting and datasets in the EmerGNN[1] method, to predict drug-drug interactions between emerging drugs and existing drugs. The experimental results are shown in Table G2. It can be seen that there is a significant performance drop from the transductive setting to the inductive setting, which shows that the DDI prediction for new drugs is more difficult. Emergnn, which is specifically designed for new drug prediction, achieves optimal performance. However, **CSSE-DDI still achieves impressive performance**, which is mainly due to the robust learning ability of NAS technology for unknown data.

|Table G2|Drugbank||TWOSIDES||
|-------|-------|-------|-------|-------|
| |F1|ACC|ROC-AUC|PR-AUC|
|CompGCN|30.98±3.26|52.76±0.46|84.83±1.02|83.68±1.86|
|Decagon|11.39±0.79|32.56±0.92|57.49±1.75|59.38±1.09|
|SumGNN|26.57±1.59|44.30±1.04|80.02±2.17|78.42±1.62|
|KnowDDI|31.14±1.24|53.44±1.73|84.23±2.63|82.58±1.94|
|EmerGNN|58.13±1.36|69.53±1.97|87.42±0.39|86.20±0.71|
|**CSSE-DDI**|37.24±1.13|58.57±0.85|88.33±0.52|86.47±0.27|

Please let us know if there are any outstanding concerns, and we are happy to discuss them. We would appreciate it if you could take our responses into consideration when making the final evaluation of our work.

Sincerely,

Authors


**References**

[1] Emerging Drug Interaction Prediction Enabled by Flow-based Graph Neural Network with Biomedical Network. Nature Computational Science' 23.

---

### Decision · Program_Chairs · 2024-09-25

**Decision:**

Accept (poster)

**Comment:**

The paper presents an approach for drug-drug interactions (DDIs) prediction, based on subgraph selection and encoding. Specifically, they use  neural architecture search (NAS) to customize the subgraph-based pipeline to the data at hand, motivated by the fact that different DDI datasets have different semantic natures of interactions.

The presented approach seems sufficiently novel, the experimental part is convincing, the use case is a nice add-on that showcases the interpretability of the approach. I recommend acceptance.